# Therapeutic Potential of CAR-CIK Cells in Acute Leukemia Relapsed Post Allogeneic Stem Cell Transplantation

**DOI:** 10.3390/cancers18010032

**Published:** 2025-12-22

**Authors:** Martina Canichella, Paolo de Fabritiis, Elisabetta Abruzzese

**Affiliations:** 1Hematology, St. Eugenio Hospital, ASL Roma2, 00144 Rome, Italy; 2Department of Biomedicina e Prevenzione, Tor Vergata University, 00133 Rome, Italy

**Keywords:** adoptive cellular therapy (ACT), donor lymphocytes infusion (DLI), cytokine-induced killer (CIK), chimeric antigen receptors (CAR)-CIK, graft-versus-leukemia (GvL)

## Abstract

Adoptive cellular therapy has long been viewed as an attractive strategy to prevent and treat post-transplant relapse in acute leukemia. Donor lymphocyte infusions (DLIs) remain the conventional approach, offering clear graft-versus-leukemia (GVL) activity but at the cost of a significant risk of graft-versus-host disease (GvHD). Cytokine-induced killer (CIK) cells have recently emerged as an alternative, retaining robust GVL effects while generally inducing less GvHD. Building on these advantages—and inspired by the clinical success of CAR T-cell therapies in B-cell malignancies—a new platform has been developed: CAR-CIK cells, which integrate CIK biology with chimeric antigen receptor engineering to enhance efficacy with reduced toxicity. This review summarizes the key clinical data on CAR-CIK therapy for managing post-transplant relapse in B-cell acute lymphoblastic leukemia and acute myeloid leukemia.

## 1. Introduction

Relapse following allogeneic hematopoietic stem cell transplantation (allo-HSCT) remains the major cause of treatment failure in acute leukemias, with particularly poor outcomes in both acute lymphoblastic leukemia (ALL) and acute myeloid leukemia (AML). In patients with B-cell ALL, post-transplant relapse is associated with 1-year overall survival ranging from approximately 17% to 44%, and 2-year survival reaching up to 35–55% in selected cohorts receiving salvage therapies [1,2,3]. Similarly, patients with AML relapsing after allo-HSCT have 1-year survival generally between 17% and 23%, with 2-year survival rarely exceeding 26%, even among younger patients or those receiving intensive salvage strategies such as donor lymphocyte infusions (DLI) or second transplantation [4,5,6,7]. These data highlight the urgent need for more effective interventions to prevent relapse and improve long-term survival in high-risk leukemia patients.

Adoptive cellular therapy (ACT) has emerged as a key strategy to address post-transplant relapse. Historically, DLI has been the cornerstone of ACT in relapse post allo-HSCT, aiming to enhance the graft-versus-leukemia (GvL) effect, particularly in patients with measurable residual disease (MRD) positivity [8,9,10,11]. However, despite its potential, DLI is limited by the risk of graft-versus-host disease (GvHD) and modest efficacy in sustaining durable antileukemic responses [12,13,14]. To overcome these limitations, alternative donor-derived cellular platforms have been explored, including cytokine-induced killer (CIK) cells. CIK cells, generated ex vivo from peripheral blood mononuclear cells with interferon-γ, anti-CD3 antibodies, and interleukin-2, exhibit a mixed T- and NK-cell phenotype (CD3^+^CD56^+^) and mediate MHC-unrestricted antileukemic activity via the NKG2D receptor [15,16,17,18,19,20,21]. Although they are generally safe with minimal GvHD risk, their clinical impact has been modest [22].

In recent decades, chimeric antigen receptor T-cell (CAR-T) therapy has become the predominant cellular immunotherapy for certain B-cell malignancies, including B-cell ALL, B-cell lymphomas, and multiple myeloma (MM). CAR-T cells are typically generated from autologous lymphocytes through viral vector transduction and are engineered to target specific antigens, such as CD19 in B-ALL and non-Hodgkin lymphomas or B-cell maturation antigen (BCMA) in MM. In acute myeloid leukemia, the most commonly investigated targets in the experimental setting include CD33 and CD146, as outlined below [23,24,25,26,27,28]. While effective, CAR-T therapy is constrained by complex manufacturing, immune-mediated toxicities, limited persistence, and risk of antigen escape [29]. To address these challenges and with the aim of improving strategies in the relapse post-allo-HSCT, CAR engineering has been applied to CIK cells, giving rise to CAR-CIK cells. This approach aims to combine the favorable safety profile and innate antitumor properties of CIK cells with the enhanced specificity and potency of CAR technology.

This review focuses on the development and clinical application of CAR-CIK cells in the post-transplant setting. Table 1 summarizes the principal differences among donor-derived cellular therapies, and Figure 1 illustrates the rationale underlying CAR-CIK design. By synthesizing current evidence, we aim to provide a comprehensive overview of CAR-CIK as a potential strategy to prevent or treat relapse after allo-HSCT.

### 1.1. Donor Lymphocyte Infusions (DLI)

DLI are adoptive immunotherapy products derived from the peripheral blood of hematopoietic stem cell transplant donors. They are typically collected via leukapheresis and consist of unmanipulated or minimally processed donor T lymphocytes intended to enhance the GvL effect in patients who have relapsed after allo-HSCT or are at high risk of relapse [8,9,10]. The biological rationale relies on the ability of donor T cells to recognize residual malignant cells in the recipient, mediating cytotoxicity and tumor clearance. However, infusion of donor lymphocytes carries a significant risk of inducing or reactivating GvHD, which represents the main dose-limiting toxicity of DLI [12,13]. More recently, interest has expanded toward using DLI in settings of measurable residual disease MRD or as prophylaxis in patients at high risk of relapse. In these contexts, DLI may maximize the GvL effect while tumor burden is low, improving the likelihood of sustained remission and possibly reducing complications [13]. According to expert recommendations from the European Society for Blood and Marrow Transplantation (EBMT), careful patient selection, dose escalation strategies, and rigorous monitoring for GVHD remain essential to balance efficacy and safety [14].

### 1.2. Cytokine-Induced Killer (CIK) Cells

CIK are a heterogeneous population of ex vivo expanded T lymphocytes characterized by a CD3^+^CD56^+^ phenotype, generated from peripheral blood mononuclear cells (PBMCs) through stimulation with IFN-γ, anti-CD3 antibodies, and IL-2 [15,16,17]. The rationale for their use relies on their potent MHC-unrestricted cytotoxicity against tumor cells, mediated through perforin, granzyme B, FasL, and NKG2D-dependent mechanisms [18,19]. CIK cells combine the proliferative capacity of T lymphocytes with the innate-like tumor recognition of NK cells, allowing rapid expansion to clinically relevant numbers. Clinically, CIK cells have been evaluated in both hematologic malignancies and solid tumors, demonstrating an overall favorable safety profile with minimal graft-versus-host disease when administered allogeneically [20,21]. Recent studies and reviews highlight ongoing efforts to optimize their antitumor activity, standardize manufacturing protocols, and explore combinatorial strategies, including CAR-engineering and combination with checkpoint inhibitors [22,23].

### 1.3. Chimeric Antigen Receptor (CAR) T Cells

CAR-T cells are genetically engineered autologous T lymphocytes designed to express a chimeric antigen receptor that combines (1) an extracellular single-chain variable fragment (scFv) targeting a tumor-associated antigen (e.g., CD19), (2) a transmembrane domain coupled with the CD3ζ signaling module, and (3) one or more co-stimulatory domains (e.g., CD28 or 4-1BB) [24]. CAR constructs are typically introduced into T cells via viral vectors (retroviral or lentiviral), although non-viral transposon-based systems such as the Sleeping Beauty (SB) platform are increasingly explored for stable gene integration, offering reduced production complexity and cost [25]. Most commercially approved CAR-T products are patient-derived, ensuring autologous compatibility and reducing the risk of alloimmune reactions. Clinically, anti-CD19 CAR-T cells have demonstrated high efficacy in patients with relapsed/refractory B-cell malignancies, including diffuse large B-cell lymphoma (DLBCL), other aggressive B-cell lymphomas, and B-cell acute lymphoblastic leukemia (B-ALL) [26,27,28,29,30]. In MM, CAR-T therapies targeting plasma-cell antigens (e.g., BCMA) have shown promising overall response rates in heavily pre-treated populations [31]. Tisagenlecleucel (tisa-cel) is an anti-CD19 CAR T-cell therapy approved for the treatment of pediatric and young adult patients up to 25 years of age with R/R B-ALL while Brexucabtagene autoleucel (brexu-cel) is approved for the treatment of adult patients with R/R mantle cell lymphoma B-ALL. The long term follow-up data from the ELIANA and ZUMA-3 trials show that both Tisagenlecleucel (in pediatric/AYA B-ALL) and Brexucabtagene autoleucel (in adult B-ALL) can induce durable remissions in a subset of patients, with survival extending beyond 3–5 years (overall remission rate 82% for ELIANA; median overall survival 25.6 months for ZUMA-3) [32,33]. Regarding safety, CAR-T therapy is associated with unique toxicities, including cytokine release syndrome (CRS) and immune effector cell-associated neurotoxicity syndrome (ICANS), which can range from mild to life-threatening. These adverse events require careful monitoring and management, often involving supportive care and targeted interventions such as tocilizumab or corticosteroids [34]. Overall, the balance of efficacy and manageable safety profile has established CAR-T therapy as a standard-of-care option for selected patients with R/R B-cell malignancies.

### 1.4. CAR-CIK Cells

CAR-CIK cells represent an emerging cellular immunotherapy platform that integrates the inherent advantages of CIK cells-including MHC-unrestricted cytotoxicity, rapid ex vivo expansion, and a favorable safety profile with minimal risk of GvHD- while integrating the targeted specificity of CAR technology (Figure 1) [35]. CAR-CIK cells are engineered to express tumor-specific CAR constructs (e.g., anti-CD19), enabling selective recognition and killing of malignant cells similar to conventional CAR-T cells. Unlike most autologous CAR-T products, CAR-CIK cells are commonly generated using non-viral gene transfer systems, particularly the SB transposon platform, which allows stable genomic integration of the CAR transgene while reducing manufacturing complexity, cost, and regulatory hurdles associated with viral vectors [36]. This hybrid approach combines the cytotoxic breadth and proliferation capacity of CIK cells with the antigen-specific precision of CAR-T therapy, providing a potentially safer and more versatile immunotherapy option. While detailed clinical and mechanistic aspects of CAR-CIK cells will be discussed in subsequent sections. Early preclinical and translational studies suggest that these cells can expand in vivo, persist for weeks to months, and mediate potent antitumor activity with a generally manageable safety profile [36,37]. As such, CAR-CIK cells are increasingly viewed as a next-generation platform capable of overcoming some limitations of conventional CAR-T therapy, particularly regarding safety, manufacturing flexibility, and applicability in allogeneic settings. Firstly, CAR-CIK cells are generated from donor-derived CIKs rather than autologous T cells and can be engineered using non-viral systems such as the Sleeping Beauty (SB) system is a non-viral gene transfer technology based on a transposon–transposase platform that enables stable integration of a target gene into the host cell genome, offering a flexible and cost-effective approach for genetic engineering of immune cells. This approach seeks to integrate the safety profile of CIK cells with the enhanced antitumor specificity of CARs, while leveraging donor availability to provide an “off-the-shelf” therapeutic option.

## 2. CAR-CIK in Acute Lymphoblastic Leukemia (ALL)

The prognosis of relapsed B-ALL after allo-HSCT remains particularly poor. Reported outcomes indicate a median overall survival (OS) of approximately 6–8 months, with 1-year OS around 40% [38,39]. In the post-transplant setting, as mentioned above, several forms of cellular therapy have been approved, offering novel therapeutic strategies for patients with relapsed or refractory disease. Donor-derived CIK cells have demonstrated encouraging results in the post-transplant setting, particularly as a strategy to prevent or treat relapse [40,41]. In 2016, Lussana et al. published the final analysis of a multicenter phase II trial evaluating CIK cells in 74 patients relapsing after allo-HSCT, including 20 cases with ALL [42]. Treatment was feasible in the vast majority of patients, with 59 patients receiving at least one infusion and 43 (58%) completing the full schedule of DLI followed by CIK cell administration. Toxicity was manageable: acute GvHD occurred in 16% of patients, mostly grades I–II, while severe grades III–IV were rare; chronic GvHD was reported in 15%, and no excess of non-GvHD severe adverse events was observed. Importantly, antileukemic activity was documented across hematologic subtypes, with an overall complete remission (CR) rate of 26% (19/73), partial remission in 4% (3/73), and disease stabilization in 11% (8/73). At one year, progression-free survival (PFS) and OS were 31% and 51%, respectively, while three-year PFS and OS were 29% and 40%. Within the ALL cohort, clinical responses were confirmed, albeit heterogeneous in durability, and some patients achieved long-lasting molecular remissions. Notably, patients treated at the stage of molecular or cytogenetic relapse derived the greatest benefit, supporting the concept that donor-derived CIK cells can mediate clinically meaningful GvL activity in B-ALL without compromising safety. In the subsequent years, the impact of CAR T-cell therapy also extended to the post-transplant setting. In particular the integration of CAR technology into CIK cells leads to the development of an innovative cellular platform designed to reinforce GvL activity while maintaining a favorable safety profile [43,44]. Specifically, in B-ALL, donor-derived CIK cells were genetically engineered to express a CD19-directed CAR using the SB transposon system. This non-viral approach provided a valuable alternative to conventional viral vector based CAR T-cell platforms, overcoming several of their limitations, including stringent regulatory requirements and the reliance on complex, time-intensive producer cell lines, which ultimately restrict product availability and increase manufacturing costs.

Notably, in 2018 Magnani et al. was the first to demonstrate preclinical efficacy and feasibility of CARCIK-CD19 generated with the SB transposon system. The study demonstrated that CARCIK cells displayed potent cytotoxic activity against CD19^+^ leukemic targets both in vitro and in xenograft models, while maintaining the hallmark low alloreactivity typical of unmodified CIK cells [45]. These findings provided the proof-of-concept for clinical translation of a non-viral, SB-based platform capable of achieving efficient and cost-effective CAR gene transfer under good manufacturing practice (GMP) conditions. The first-in-human experience was reported by Magnani and colleagues in the phase I/II FT01CARCIK trial (NCT03389035), which enrolled 21 patients with R/R B-ALL, the majority of whom relapsed after allo-HSCT [45]. In this heavily pretreated population, 13 patients (62%) achieved a CR or CR with incomplete hematologic recovery (CRi) following infusion of donor-derived CARCIK-CD19 cells. Importantly, the therapy was associated with a highly favorable safety profile: CRS occurred only in a minority of cases and was limited to grade I-II, no neurotoxicity was observed, and, notably, no severe GvHD was reported, underscoring the intrinsic safety advantage of the CIK platform.

Lussana et al. investigated the use of donor-derived CARCIK-CD19 cells engineered via the Sleeping Beauty transposon system in patients with B-cell acute lymphoblastic leukemia who experienced relapse following allogeneic hematopoietic stem cell transplantation [46]. A total of 36 patients, including 4 children and 32 adults, received the recommended dose of more than 7.5 × 10^6^ transduced CARCIK-CD19 cells per kilogram. Treatment was administered within three settings: 15 patients in the FT01CARCIK study, six under a compassionate use program approved by the Italian Medicines Agency (FT02CARCIK), and 15 in the phase 2 FT03CARCIK study. Complete remission was observed in 30 patients (83.3%), with 89% of responders achieving minimal residual disease negativity. Safety outcomes included grade ≤ 2 cytokine release syndrome in 15 patients, grade 2 ICANS in one patient, and late-onset grade 3 peripheral neurotoxicity in two patients, with no cases of graft-versus-host disease reported. At a median follow-up of 2.2 years, 1-year overall survival was 57% and 1-year event-free survival was 32%. CARCIK cells demonstrated rapid in vivo expansion and persisted for over 2 years, exhibiting high clonal diversity. Building on these data, preclinical refinements have explored the “armoring” of CARCIK cells with cytokines such as interleukin-18, as reported by Debenedette and colleagues, which enhanced expansion and persistence in xenograft models and translated into superior antitumor activity [47]. At the clinical level, a second-generation academic trial (NCT05869279) is currently underway, expanding the evaluation of CARCIK-CD19 to a broader cohort of patients with R/R B-cell malignancies, including both B-ALL and non-Hodgkin lymphomas. Taken together, these studies suggest that CARCIK-CD19 cells represent a promising allogeneic, off-the-shelf alternative to conventional CAR-T and donor-T cells, combining potent antileukemic efficacy with a markedly reduced incidence of treatment-related toxicities (Table 2).

## 3. CAR-CIK in Acute Myeloid Leukemia

The prognosis of AML relapsed after allo-HSCT is particularly dismal with median OS less than 6 months and 1 year-OS around 10–15% [4,48]. Therefore, the development of strategies to avoid or treat relapse is a challenge. From this point of view, CARCIK platform has also been investigated also in the setting of AML relapsed post allo-HSCT. Although the first investigations confirmed that donor-derived CIK cells could exert GvL activity in AML with an excellent safety profile the clinical efficacy remained modest. This finding is partially related to the intrinsic resistance of myeloid blasts and the protective influence of the bone marrow niche. To overcome these barriers, researchers introduced a CD33-specific CAR into CIK cells, which markedly enhanced their effector functions: in vitro assays demonstrated cytotoxicity rates of approximately 60–70% against CD33-positive AML cell lines at conventional effector-to-target ratios, significantly outperforming unmodified CIKs [49]. Building on this approach, tandem CAR constructs targeting both CD33 on leukemic blast and CD146 widely expressed on mesenchymal stromal cells were developed, aiming not only to eradicate leukemic blasts but also to disrupt the supportive vascular and stromal components of the marrow microenvironment [50]. Although primary mesenchymal stromal cells continued to exert partial inhibitory effects in co-culture systems, these dual CAR-CIKs exhibited superior capacity to overcome microenvironment-mediated resistance compared to single-target CARs. The most recent advance came from Biondi and colleagues in 2023, who combined CD33-directed CARs with enforced CXCR4 expression to enhance bone marrow tropism [51]. In xenograft models, these modified CARCIKs selectively homed to the marrow, resulting in significantly improved leukemia control and a median survival of ~110 days, compared with 57.5 days in untreated controls and 77–87 days in mice receiving conventional CAR-CIKs. Table 3 summarizes the main applications of CAR-CIK cells in AML.

## 4. Discussion

For patients with high-risk ALL and AML undergoing allo-HSCT, defining effective post-transplant strategies remains essential to prevent relapse. DLI have traditionally represented a consolidated option, although their efficacy is counterbalanced by a substantial risk of severe GvHD. To enhance the GvL effect while reducing toxicity, CIK cells have been explored with encouraging results, and on this basis CARCIK cells have been developed as a next step to further improve post-transplant disease control. The evolution of CARCIK therapy currently revolves around several interconnected aspects. A first crucial consideration concerns target selection. As observed for CAR-T cells, the efficacy of CARCIK cells depends on the presence of leukemia-specific and stable antigens. Their activity appears particularly promising in lymphoid malignancies, whereas in AML their role remains limited, mainly due to the lack of suitable targets expressed on leukemic blasts but absent on normal hematopoietic cells, as well as the protective influence of the leukemic microenvironment [52,53]. These factors significantly restrict the broad applicability of engineered immune therapies in myeloid diseases, and although novel approaches—such as dual or tandem CAR constructs—are being investigated, a definitive solution has not yet been identified. Another key aspect relates to the choice of the immune effector platform. Compared with conventional autologous CAR-T cells, CARCIK cells offer distinctive advantages: being donor-derived, they avoid the functional exhaustion commonly observed in autologous products generated from heavily pretreated patients, and their engineering based on the SB transposon system provides a non-viral, cost-effective, and logistically advantageous gene-transfer method. From a safety standpoint, CARCIK cells also display a lower incidence of GvHD compared with DLI, reinforcing their potential as an allogeneic cellular therapy platform. Equally important is the question of timing. The optimal moment for administering CARCIK cells after allo-HSCT remains under investigation, yet emerging evidence suggests that their greatest therapeutic impact may lie before the onset of overt hematologic relapse. In a structured post-transplant monitoring strategy, the most promising window could therefore be the pre-emptive phase, particularly at the time of MRD conversion from negativity to positivity. In selected high-risk patients, their use might even be considered prophylactically or as a maintenance approach, to prevent molecular progression and stabilize remission. Defining this early intervention window will be essential to fully harness the therapeutic potential of CARCIK therapy. Overall, CARCIK cells represent a promising platform for post-transplant immune modulation, particularly in lymphoid malignancies. However, in AML their clinical development continues to be hindered by biological limitations similar to those encountered with CAR-T therapies, underscoring the need for further research into target identification and microenvironment modulation. In this context, an international panel of experts has recently proposed consensus recommendations aimed at harmonizing trial design, reporting standards, and future development strategies for AML-directed CAR-T therapies, which may also be informative for the advancement of CARCIK approaches [54].

## 5. Conclusions

CARCIK represents a novel and promising therapeutic strategy particularly in the management of B-ALL. Nonetheless, several issues still need to be addressed and overcome in order to fully realize their potential.

In the next future the clinical application of CARCIK could be expanded. Ongoing studies are evaluating its role in the treatment of lymphomas, with promising preliminary results. Furthermore, there is increasing interest in extending this approach to non-hematologic malignancies, where CARCIK strategies may provide novel opportunities for therapeutic innovation.

## Figures and Tables

**Figure 1 cancers-18-00032-f001:**
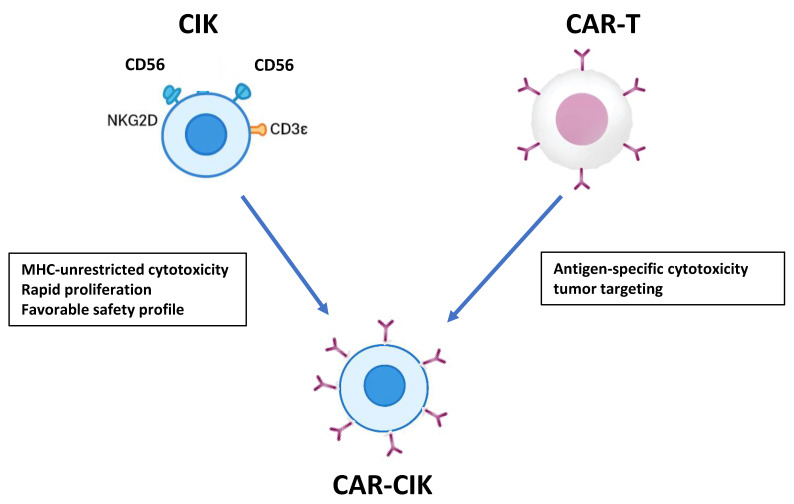
CARCIK: rationale and structure.

**Table 1 cancers-18-00032-t001:** Comparative Overview of CAR-T, CAR-CIK, and DLI in the Post-HSCT.

Feature	DLI	CIK	CAR-CIK
Source	Donor T cells	Donor T/NK-like cells expanded ex vivo	Donor CIK cells genetically engineered with CAR
Antileukemic effect	GvL effect, moderate	Broader GvL effect, NK-like cytotoxicity	Potent, antigen-specific cytotoxicity plus NK-like activity
Alloreactivity/GvHD risk	High	Low to moderate	Low (maintained safety of CIK platform)
Target specificity	Non-specific	Limited	CAR-directed, antigen-specific (e.g., CD19, CD33)
Persistence	Short to moderate	Moderate	Improved persistence depending on CAR and modifications (e.g., CXCR4)
Microenvironmental targeting	None	Limited	Can be engineered (e.g., CD33xCD146 tandem CAR for AML niche targeting)
Manufacturing complexity	Minimal	Moderate (ex vivo expansion)	High (requires CAR transduction, possibly transposon or viral vectors)
Clinical use post-transplant	Relapse treatment, prophylaxis limited	Relapse prevention/treatment	Relapse prevention/treatment, potentially safer than CAR-T

HSCT: Hematopoietic stem cell transplantation; DLI: Donor Lymphocyte Infusion; CIK: Cytokine-Induced Killer Cells; CAR-CIK: Chimeric antigen receptor CIK Cells; GvL: graft versus leukemia.

**Table 2 cancers-18-00032-t002:** Main Clinical Trials of CAR-CIK in ALL.

Type of Study	Model/Population	CAR Target	Findings	Safety/GvHD	Ref.
Phase I/II CARCIK-SB (early experience reported)	13 (4 pediatric, 9 adult)	CD19-CAR	CR/CRi in 6/7 at highest dose; robust expansion; CAR detectable up to 10 months	2 CRS G1, 1 × CRS G2; no GVHD reported	[45]
CARCIK (NCT03389035/FT01/FT02/FT03)	36 (4 pediatric, 32 adult)	CD19-CAR	Durable response; 1-year OS 57%, 1-year EFS 32%; 83% ORR, 86% MRD-neg CR	Mostly low-grade toxicity; no GVHD observed; 15/36 had CRS ≤ G2; ICANS G2 in 1; late peripheral neurotox G3 in 2	[46]

ALL: acute lymphoblastic leukemia; GvHD: graft versus host disease; CIK: Cytokine-Induced Killer Cells; CAR-CIK: Chimeric antigen receptor CIK Cells; CR: complete remission; MRD: measurable residual disease; OS: overall survival; EFS: event-free-survival; CRS: cytokine release syndrome; G: grade.

**Table 3 cancers-18-00032-t003:** Main Clinical Trials of CAR-CIK in AML.

Type of Study	Model/Population	CAR Target	Findings	Ref.
preclinical	In vitro AML + stromal co-culture	Tandem CD33 × CD146	Demonstrates microenvironment targeting, dual CAR approach	[50]
preclinical	AML xenograft murine models	CD33-CAR + CXCR4	Highlights importance of homing and niche targeting for durable effect	[51]

AML: acute myeloid leukemia; CAR-CIK: Chimeric antigen receptor CIK Cells.

## Data Availability

No new data were created or analyzed in this study. Data sharing is not applicable to this article.

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
