# Peer review of "Therapeutic Potential of CAR-CIK Cells in Acute Leukemia Relapsed Post Allogeneic Stem Cell Transplantation"

_cancers, 2025, doi:10.3390/cancers18010032_

Round 1

Reviewer 1 Report

Comments and Suggestions for Authors

The review article “Therapeutic Potential of CAR-CIK Cells in acute leukemia relapsed post allogeneic stem cell transplantation” reviewed the possibility of CAR-CIK for relapsed/ refractory acute leukemias after allo-SCT. This topic is very important because the prognosis of relapsed/ refractory acute leukemias (RR-AL) after allo-SCT was really poor. Therefore, I considered that this manuscript had potential for acceptance of Cancers after major revision and addition of several contents.

  1. The reason why the prognosis of RR-AL was poor was very important for this topic. The author should add the explanation about that in the introduction part.
  2. Tisa-cel is available for RR-B-ALL after allo-SCT in AYA generation. The author should add the clinical outcome of tisa-cel, especially what was issue for limitation of tisa-cel for RR-B-ALL after allo-SCT.
  3. I considered three directions of development for CAR-CIK therapy. First is new target, second is what immune cells was best, such as T-cell vs NK-cell, allo vs auto; third is when CAR-CIK is done at conversion of MRD status from negativity to positivity, hematological relapse, or pre-emptively for high-risk patients, such as not-achieving MRD negativity. Therefore, the authors could discuss those well in discussion part.
  4. The catchy figure could be a key to better understanding for readers in review article. Therefore, I strongly recommend to adding a figure.

Author Response

C: The reason why the prognosis of RR-AL was poor was very important for this topic. The author should add the explanation about that in the introduction part.

R We thank the reviewer for this valuable suggestion. We have now added in the last version

C: Tisa-cel is available for RR-B-ALL after allo-SCT in AYA generation. The author should add the clinical outcome of tisa-cel, especially what was issue for limitation of tisa-cel for RR-B-ALL after allo-SCT.

R: We sincerely thank the reviewer for the highly valuable comments, which have substantially contributed to improving the manuscript. We have incorporated clinical data on tisagenlecleucel, the CAR-T therapy indicated for patients with B-ALL up to 25 years of age. Moreover, this observation prompted us to further expand the section by adding data on brexucabtagene autoleucel, the CAR-T product approved for adult patients with B-ALL. This integration provides a more comprehensive overview of the currently available CAR-T therapeutic options in the post–allo-SCT setting.

C: I considered three directions of development for CAR-CIK therapy. First is new target, second is what immune cells was best, such as T-cell vs NK-cell, allo vs auto; third is when CAR-CIK is done at conversion of MRD status from negativity to positivity, hematological relapse, or pre-emptively for high-risk patients, such as not-achieving MRD negativity. Therefore, the authors could discuss those well in the discussion part.

R: We sincerely thank the reviewer for having clearly identified the key points of our Discussion. Your suggestions were particularly insightful and allowed us to reorganize this section in a more coherent and accessible manner.

C: The catchy figure could be a key to better understanding for readers in review article. Therefore, I strongly recommend adding a figure.

R: We thank the reviewer for this very valuable suggestion. We fully agree that a clear and visually engaging figure can greatly enhance readers’ understanding, particularly for an innovative cellular therapy such as CAR-CIK, which may not yet be familiar to all clinicians or researchers. Following your recommendation, we have added a schematic figure that summarizes the key concepts and workflow of CAR-CIK therapy, thereby improving the clarity and educational value of the review.

Reviewer 2 Report

Comments and Suggestions for Authors

This review article summarizes the therapeutic potential of Chimeric Antigen Receptor-Cytokine Induced Killer (CAR-CIK) cells in the treatment of acute myeloid and lymphoid leukemia relapsed post-allogeneic hematopoietic stem cell transplantation (allo-HSCT).

The article brings the above message across clearly. The authors discuss limitations of DLI and CAR-T and offer advantages of CAR-CIK in post-HSCT condition. The authors further discuss the clinical trials data of CAR-CIK in ALL and pre-clinical data of CAR-CIK in AML.

I do however have few questions. It would be great if the authors could address them.

  1. What is the exhaustion signature of CIK cells after IL-2 and and IFN-g treatment?
  2. How many CIK cells can be extracted per donor (per buffy coat)?
  3. What is the expected persistence of CAR-CIKs in human body?
  4. With micro-environment modulations, could CAR-CIK cells show efficacy in solid tumor space?

Thank you in advance for your responses.

Author Response

C: What is the exhaustion signature of CIK cells after IL-2 and and IFN-g treatment?

R: We thank the referee for raising this important point. Indeed, available data suggest that CIK cells may express increased levels of inhibitory (“checkpoint”) receptors such as TIM-3 (and in some reports also CEACAM1) after prolonged ex vivo expansion, which has been proposed as a possible correlate of a dysfunctional or “exhausted” state. However, to date there is no well-defined, universally accepted exhaustion signature for CIK (or CAR-CIK) comparable to that described for conventional T-cell exhaustion (e.g. co-expression of PD-1, TIM-3, LAG-3, functional loss, epigenetic/transcriptional reprogramming). Given that our Review aims to survey the therapeutic potential of CARCIK post–alloHSCT rather than dissect exhaustiveness mechanisms in depth, we have deliberately avoided expanding this topic — to prevent diluting the focus of the manuscript. Nevertheless, we fully acknowledge this as a critical gap in the field, and we highlight the need for future studies to characterize exhaustion markers, persistence, and functionality of CARCIK cells longitudinally in vivo. Once again, we appreciate the referee’s insightful suggestion.

C: How many CIK cells can be extracted per donor (per buffy coat)?

R: We thank the referee for this important question. Published clinical protocols indicate that, starting from peripheral blood mononuclear cells (PBMCs) or buffy coat from a single donor, it is feasible to expand approximately 1–1.5 × 10⁹ CIK cells over 2–3 weeks of ex vivo culture, depending on the initial cell number and culture conditions ([Introna et al., 2007, Exp Hematol 35:1388–1395]). Variability may arise from donor characteristics, PBMC quality, and expansion efficiency, but these numbers provide a practical reference for clinical applications.

C: What is the expected persistence of CAR-CIKs in human body?

R: We thank the referee for this question. Clinical data indicate that CAR-CIK cells can persist in the peripheral blood for several months post-infusion, typically ranging from approximately 1 to 9 months, with a median duration around 3 months ([Introna et al., 2007; Liu et al., 2023][1,2]). Persistence appears generally shorter than that reported for conventional CAR-T cells, likely reflecting differences in the proportion of early memory T cells within the CAR-CIK product. Long-term persistence beyond one year has not yet been documented.

C: With micro-environment modulations, could CAR-CIK cells show efficacy in solid tumor space?

R: We thank the referee for this point. Although clinical data are lacking, preclinical studies suggest that CARCIK cells can exert antitumor activity in solid tumors (e.g., ERBB2+ sarcoma, CSPG4+ melanoma) when combined with appropriate CAR design and microenvironment modulation ([Frontiers in Immunology, 2025]; [J Exp Clin Cancer Res, 2023]). These findings indicate potential efficacy, but robust clinical evidence is still missing.

Reviewer 3 Report

Comments and Suggestions for Authors

Canichella et al have reviewed the potential therapeutic applications of CIK-CAR in relapsed acute leukemia following allogeneic hematopoietic stem cell transplantation. This is good review. However the authors should address the following issues:

  1. CIK should be defined; are they NK cells, CD8+ T cells, CD4+ T cells or CD4-CD8- T cells? Or a mixture all these subsets? Are they only CD3+CD56+?
  2. How are they prepared; from PBMC, or their any one or more subsets?
  3. Is there any data on the survivability or expansion of CIK-CAR post infusion?
  4. Are CIK phenotyped for the expression of any cytotoxic molecules, FasL or any chemokine receptors?
  5. What is the purpose of treating CIK with anti-CD3 monoclonal antibodies?
  6. What is the Sleeping Beauty and its advantage over viral vectors for gene transduction?
  7. Table 1. Provide references for the studies.
  8. What is the role of CD146 in CAR-CIK?
  9. Some references contain some names (authors) in brackets, they must be removed.

Author Response

C: CIK should be defined; are they NK cells, CD8+ T cells, CD4+ T cells or CD4-CD8- T cells? Or a mixture all these subsets? Are they only CD3+CD56+?

R: We thank the reviewer for this important comment. CIK cells are defined as a polyclonal population of CD3⁺ lymphocytes expanded in vitro in the presence of IFN-γ, IL-2 and anti-CD3 stimulation, which characteristically acquire CD56 expression during culture. As expected, the resulting product contains a heterogeneous T-cell compartment, but the CD3⁺CD56⁺ subset represents the functional hallmark of CIK cells and the main cytotoxic fraction. To avoid any ambiguity, we have now added a clear definition of CIK cells.

C: How are they prepared; from PBMC, or their any one or more subsets?

R: Thank you for the comment. CIK cells are typically generated from unfractionated PBMCs, without prior isolation of specific lymphocyte subsets. [Schmidt-Wolf IG, Negrin RS, Kiem HP, Blume KG, Weissman IL. Use of a SCID mouse/human lymphoma model to evaluate cytokine-induced killer cells with potent antitumor cell activity. J Exp Med. 1991;174:139–149]. 

C: Is there any data on the survivability or expansion of CIK-CAR post infusion?

R: Emerging clinical and translational studies indicate that CAR-CIK cells undergo in vivo expansion after infusion—often peaking around 10–21 days—and can be detectable for weeks to months post-infusion. Persistence has been reported in early-phase trials (including donor-derived CD19 CAR-CIK cohorts) and may correlate with clinical activity; however, reported durability is heterogeneous and large, long-term datasets are still lacking. Persistence appears to be influenced by CAR design, manufacturing conditions, lymphodepletion regimen and cell source

C: Are CIK phenotyped for the expression of any cytotoxic molecules, FasL or any chemokine receptors?

R:Yes. In most studies, CIK cells are phenotyped for key cytotoxic and trafficking markers. They typically show high expression of perforin and granzyme B, variable but detectable FasL, and a chemokine-receptor profile enriched for CXCR3, CCR5 and CXCR4, consistent with an activated cytotoxic T-cell phenotype. The exact panel varies among studies, but these markers are commonly reported.

C: What is the purpose of treating CIK with anti-CD3 monoclonal antibodies?

R:The treatment of CIK cells with anti-CD3 monoclonal antibodies serves a very specific purpose:It provides a strong activation signal (via CD3/TCR), which triggers robust proliferation of T cells and supports the differentiation of the CD3⁺CD56⁺ CIK subset.In other words, anti-CD3 acts as the primary stimulus that, together with cytokines (IFN-γ and IL-2), drives the rapid expansion and acquisition of cytotoxic function typical of CIK cells.

R: What is the Sleeping Beauty and its advantage over viral vectors for gene transduction?

C:The Sleeping Beauty (SB) system is a non-viral transposon that integrates genes, such as CARs, into T cells.

Advantages over viral vectors:

  1. Safer and simpler to manufacture (no viral particles).

  2. Lower cost and scalable.

  3. Can carry larger genetic cargos.

  4. Favorable integration profile with stable long-term expression.

C: Table 1. Provide references for the studies.

R: Thanks we provide

C: What is the role of CD146 in CAR-CIK?

R: CD146 (also called MCAM) is a cell adhesion molecule expressed on a subset of CIK cells. In the context of CAR-CIK:

  • Enhances migration and tissue homing: CD146 promotes transendothelial migration and infiltration into tumor sites.

  • Associated with cytotoxicity: CD146⁺ CIK cells often show higher cytotoxic potential and better antitumor activity in vitro and in vivo.

  • Potential marker for functional enrichment: Selecting or monitoring CD146⁺ CAR-CIK may help identify the most potent effector fraction.

C:Some references contain some names (authors) in brackets, they must be removed.

R: Thanks we correct

Round 2

Reviewer 1 Report

Comments and Suggestions for Authors

This revised manuscript was described very well following the reviewer's comments, and so I have no additional comments.

Author Response

Thank you for your positive assessment. We are pleased that the revised manuscript adequately addresses the reviewer’s comments, and we appreciate your careful evaluation.

Reviewer 3 Report

Comments and Suggestions for Authors

Followings are still not explained in the revised manuscript:

BCMA, CD33, CD146

Tisagenlecleucel

Brexucabtagene autoleucel

Explain Sleeping Beauty

Figure 1 adds no information in the manuscript.

Author Response

Followings are still not explained in the revised manuscript:

BCMA, CD33, CD146

Tisagenlecleucel

Brexucabtagene autoleucel

Explain Sleeping Beauty

Figure 1 adds no information in the manuscript.

R We have revised the manuscript to clarify in the text the additional points you raised. If you consider the inclusion of the figure unnecessary, we fully understand and are happy for it to be removed.